# Ataxia telangiectasia and Rad3-related inhibition by AZD6738 enhances gemcitabine-induced cytotoxic effects in bladder cancer cells

**Makoto Isono** [ORCID]**\*, Kazuki Okubo, Takako Asano, Akinori Sato**

Department of Urology, National Defense Medical College, Tokorozawa, Japan

\* mktisn@hotmail.com

**Data Availability Statement:** All relevant data are within the paper and its Supporting Information files.

## Abstract

The ataxia telangiectasia and rad3-related-checkpoint kinase 1 (ATR-CHK1) pathway is involved in DNA damage responses in many cancer cells. ATR inhibitors have been used in clinical trials in combination with radiation or chemotherapeutics; however, their effects against bladder cancer remain unclear. Here, the efficacy of combining gemcitabine with the novel ATR inhibitor AZD6738 was investigated *in vitro* in three bladder cancer cell lines (J82, T24, and UM-UC-3 cells). The effects of gemcitabine and AZD6738 on cell viability, clonogenicity, cell cycle, and apoptosis were examined. The combined use of gemcitabine and AZD6738 inhibited the viability and colony formation of bladder cancer cells compared to either treatment alone. Gemcitabine (5 nM) and AZD6738 (1 µM) inhibited cell cycle progression, causing cell accumulation in the S phase. Moreover, combined treatment enhanced cleaved poly[ADP-ribose]-polymerase expression alongside the number of annexin V-positive cells, indicating apoptosis induction. Mechanistic investigations showed that AZD6738 treatment inhibited the repair of gemcitabine-induced double-strand breaks by interfering with CHK1. Combining AZD6738 with gemcitabine could therefore be useful for bladder cancer therapy.

## Introduction

Bladder cancer has become one of the most recurrent malignant tumors affecting many patients worldwide. Nearly 30% of newly diagnosed patients present with muscle-invasive bladder cancer, and approximately 50% progress to distant metastases [1]. Over the past two decades, chemotherapy for invasive bladder cancer has been based on combinations of cisplatin and other cytotoxic drugs [1, 2]. This treatment is moderately efficacious but is limited because of the frequent development of resistance and toxicity. More than 50% of patients with bladder cancer are ineligible for cisplatin because of renal dysfunction, poor performance status, or comorbidities [3]. Novel second-line immunotherapeutic drugs such as atezolizumab have not yielded significant benefits so far, with a median overall survival of approximately 7.9 months after treatment [4]. Currently, there are no curative therapeutic options available for

**Funding:** The authors received no specific funding for this work.

**Competing interests:** The authors have declared that no competing interests exist.

patients with metastatic bladder cancer, therefore necessitating further studies regarding more effective regimens.

Cells exposed to genotoxic stress through agents such as chemotherapy undergo many different mechanisms to preserve the genomic code [5]. These include checkpoint signaling, which causes cell cycle arrest and provides time for DNA repair before cells with DNA damage enter mitosis. Ataxia telangiectasia mutated (ATM) is transiently activated upon DNA double-strand breaks (DSBs), whereas the presence of single-stranded DNA or resected DSBs recruits and activates ataxia telangiectasia and Rad3-related (ATR) [6]. ATR phosphorylates the downstream serine/threonine-specific protein checkpoint kinase 1 (CHK1), thereby preventing downstream effectors from activating cyclin-dependent kinases that promote cell cycle transition [7, 8].

Cancer cells, especially those in invasive urothelial carcinoma, the major histological subtype of bladder cancer, demonstrate increased genomic instability. One aspect of the DNA damage response (DDR) of cancer cells that differs from that of normal cells is that most cancer cells have lost one or more DDR pathways, resulting in a greater dependency on the remaining DDR pathways for survival [9]. This provides the potential for inhibitor activity that targets the DDR pathway, because the loss of one or more DDR pathways can leave cancer cells vulnerable to inhibition of the remaining pathways, inducing cancer-specific cell death. Our previous analysis revealed that drugs targeting CHKs in combination with gemcitabine efficiently inhibited cell proliferation and caused cell death in bladder cancer cells [10].

As mentioned above, the initial activation of a DDR response to replication stress starts with the recruitment of ATR, which prevents replication fork collapse and the generation of DSBs through multiple mechanisms [11]. We have postulated that the inhibition of ATR abrogates DNA damage-induced cell responses, allowing cells to enter mitosis despite DNA damage, which can lead to cell death. Despite their efficacy, however, these compounds do not seem optimal for the treatment of bladder cancer on their own, because cell death only partly occurs through apoptosis [12]. The ATR inhibitor AZD6738, with DNA-damaging agents, induces a canonical apoptotic response in several cancer types [13–16]. Gemcitabine, a nucleoside analog of deoxycytidine, has been widely used as a standard of care in several cancer types over the last 15 years. It reportedly replaces cytidine during DNA replication and arrests tumor growth, resulting in apoptosis [17, 18]. Nevertheless, the extent to which AZD6738 promotes gemcitabine-induced tumor cell death in bladder cancer cell lines remains unknown.

In this study, we investigated whether co-administration of AZD6738 influences the cytotoxic effects of gemcitabine in bladder cancer cells and examined the possible underlying mechanisms.

## Materials and methods

### Cell culture and agents

Human bladder cancer J82, T24, and UM-UC-3 cell lines were purchased from the American Type Culture Collection (Rockville, MD, USA) and were maintained in DMEM supplemented with 10% fetal bovine serum and 1% penicillin/streptomycin (Invitrogen, Carlsbad, CA, USA) at 37˚C in a humidified atmosphere of 5% $CO_2$ [19]. Gemcitabine and AZD6738 were purchased from Selleck Chemicals (Houston, TX, USA). They were dissolved in dimethyl sulfoxide (DMSO) and stored at -80˚C until use.

### Cell proliferation assay

The cancer cells were seeded in a culture medium on 96-well plates at a density of $3 \times 10^3$ cells/well and incubated at 37˚C for 24 h. Cells were treated with various concentrations of gemcitabine and/or AZD6738 for 48 h. The number of viable cells was evaluated using the 3-

(4,5-dimethylthiazol-2-yl)-5-(3-carboxymethoxyphenyl)-2-(4-sulfophenyl)-2H-tetrazolium (MTS) assay using the CellTiter 96 Aqueous kit (Promega, Madison, WI, USA). Data are expressed as the percentage of viable cells relative to the controls. The experiments were carried out in triplicate, and the data are expressed as the mean ± standard deviation (SD) of relative cell viability. A combination index (CI) analysis using the Chou-Talalay method (CalcuSyn software, Biosoft, Cambridge, UK) [20] was used to quantitatively measure the extent of the drug interaction. A CI of less than, equal to, and more than one indicates synergy, additivity, and antagonism, respectively.

## Colony-formation assay

For colony formation assays, cells were seeded into 6-well plates at a density of 100 cells per well, allowed to attach for 24 h, and then treated with 5 nM gemcitabine and/or 1 μM AZD6738 for 24 or 48 h. After 10–15 days, the cells were fixed in methanol and stained with Giemsa solution (Muto, Tokyo, Japan). Absorbance was measured at 560 nm wavelength.

## Flow cytometry

Cell cycle analyses were performed 24 and 48 h after treatment with the indicated concentrations of gemcitabine with or without AZD6738 to evaluate changes in cell cycle distribution. They were then washed with PBS and harvested through trypsinization. Harvested cells were resuspended in citrate buffer and stained with 50 μg/mL propidium iodide for 30 min at room temperature.

Annexin V assay was performed to assess apoptotic cell death and necrosis. Bladder cancer cells were stained with annexin V and 7-amino-actinomycin D (7-AAD) (Beckman Coulter, Marseille, France) 48 h after treatment with the indicated concentrations of gemcitabine and/ or AZD6738. Flow cytometry and cell sorting were performed using a FACSCalibur cell sorter (BD Biosciences, San Jose, CA, USA).

## Western blot analysis

Total protein lysates were obtained using RIPA buffer containing 150 mM NaCl, 1% Triton X-100, 0.5% deoxycholate, 1% Nonidet P-40, 0.1% sodium dodecyl sulfate (SDS), 1 mM EDTA, 50 mM Tris (pH 7.6), and 10 μL/mL protease inhibitor cocktail (Sigma Aldrich, St. Louis, MO, USA). Equal amounts of protein from each sample were separated on SDS-PAGE gels and then transferred to nitrocellulose membranes. Membranes were blocked with 5% non-fat milk or 5% bovine serum albumin in TBS-T (150 mM NaCl, 10 mM Tris, pH 7.4, and 0.1% Tween-20), and incubated with the following primary antibodies at 4°C overnight: poly [ADP-ribose] polymerase (PARP), cleaved PARP, CHK1, phosphorylated CHK1 (Ser345), pH2A.X, and Rad51 (1:1000; Cell Signaling Technology, Danvers, MA, USA), anti-CDK 4, anti-cyclin A, -B1, -D1, -E, p21$^{CIP1}$, and cdc25A (1:250; Santa Cruz Biotechnology, Santa Cruz, CA, USA), active caspase 3 (1:500; Epitomics, Burlingame, CA), and anti-actin (1:3000; Millipore, Billerica, MA, USA) as a loading control. After several washes with TBS-T, the membranes were incubated with HRP-conjugated secondary antibody (1:5000; Bio-Rad, Hercules, CA, USA) for 1 h at room temperature. The bands were detected using an enhanced chemiluminescence detection system (GE Healthcare, Wauwatosa, WI, USA).

## Statistical analysis

The statistical significance of observed differences between samples was evaluated using the Mann-Whitney U test (JMP Pro14 software, SAS Institute, Cary, NC, USA), and differences for which $p < 0.05$ were considered statistically significant.

## Results

### Inhibition of bladder cancer cell proliferation

To investigate whether AZD6738 enhances the cytotoxic effects of gemcitabine, we initially identified potential mechanisms involved in the inhibition of proliferation and clonogenicity in bladder cancer cells. In the MTS assays, AZD6738 enhanced the effects of gemcitabine on the viability of three different bladder cancer cell lines; nevertheless, up to 1.5 μM AZD6738 had little effect on its own (Fig 1A). In addition, the CI value demonstrated that the combined effect of gemcitabine and AZD6738 on the viability of bladder cancer cells was synergistic (CI < 1) under all treatment conditions (Table 1).

Similarly, the gemcitabine and AZD6738 combination treatment inhibited colony formation in the three bladder cancer cell lines, whereas gemcitabine or AZD6738 alone moderately inhibited colony formation (Fig 1B, S1 Fig), indicating that the combination treatment inhibited the long-term growth of bladder cancer cells *in vitro*.

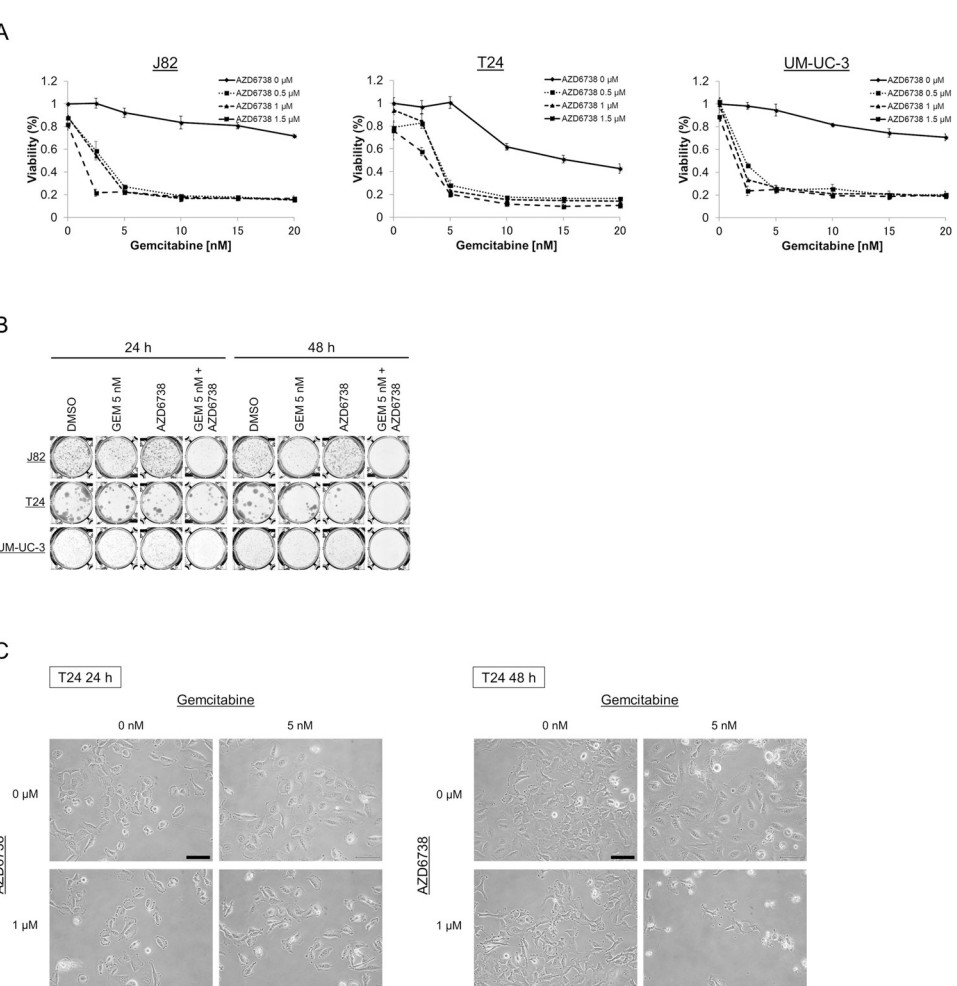

**Fig 1. Viability and clonogenicity of bladder cancer cells after treatment with gemcitabine and/or AZD6738. (A)** Relative viability in bladder cancer cells (J82, T24, and UM-UM-3) was measured using the MTS assay (mean ±SD, n = 4) after 48 h of treatment with gemcitabine and/or AZD6738. **(B)** Clonogenicity assay following 24 and 48 h of treatment with gemcitabine, AZD6738, or both compounds compared to DMSO solvent control. GEM stands for gemcitabine. The concentration of AZD6738 is 1 μM. **(C)** Photomicrographs showing characteristic morphological changes in T24 cells treated with gemcitabine and AZD6738 (24 and 48 h). Scale bar = 100 μm.

**Table 1. Combination indices.** Combination indices (CIs) calculated for the combination of gemcitabine and AZD6738 in bladder cancer cells (CI<1 indicates synergy).

| | AZD6738 (µM) | | |
| --- | --- | --- | --- |
| **Gemcitabine (nM)** | **0.2** | **0.5** | **1** |
| **J82** | | | |
| 5 | 0.149 | 0.135 | 0.135 |
| 10 | 0.251 | 0.235 | 0.243 |
| **T24** | | | |
| 5 | 0.194 | 0.177 | 0.165 |
| 10 | 0.309 | 0.291 | 0.258 |
| **UM-UC-3** | | | |
| 5 | 0.146 | 0.157 | 0.157 |
| 10 | 0.297 | 0.272 | 0.266 |

We further investigated changes in cell morphology upon gemcitabine and/or AZD6738 treatment using light microscopy (Fig 1C, S2 Fig). The number of detached, shrunken, and blebbing cells suggestive of apoptosis induction, as well as the number of attached, enlarged, and vacant-looking cells suggestive of senescent and/or necrotic cells were increased after 48 h of the combination treatment with gemcitabine and AZD6738 (Fig 1C).

## Cell cycle disturbances induced by the combination of gemcitabine and AZD6738

To follow the induction of growth arrest by gemcitabine and AZD6738, we analyzed cell cycle distribution in response to treatment (Fig 2A, S3 Fig). Gemcitabine alone and the combination treatment for 24 h increased the number of bladder cancer cells in the S phase DNA content. In T24 cells, this effect appeared to subside after 48 h of gemcitabine treatment. In the other two cell lines, cancer cells accumulated in the S phase. Conversely, the combined treatment of

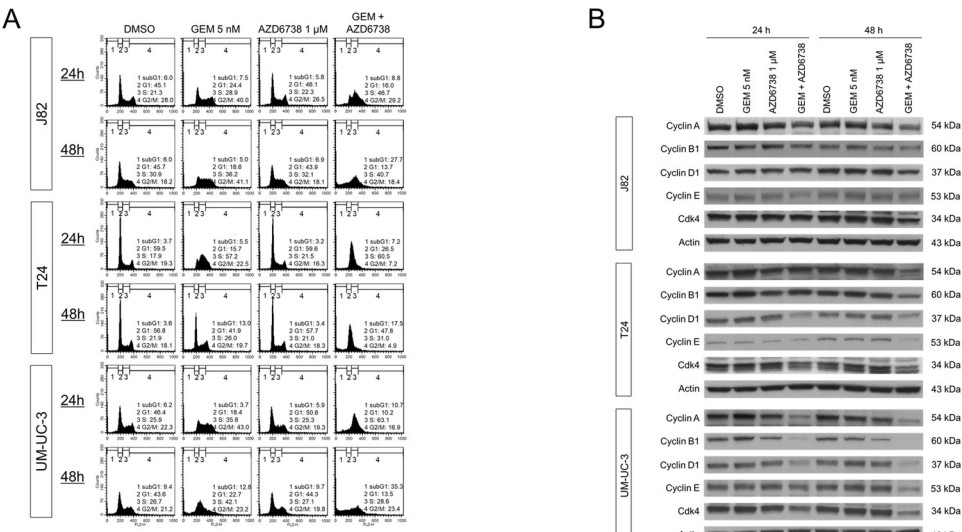

**Fig 2. Effects of gemcitabine and AZD6738 on cell cycle distribution. (A)** Flow cytometric cell cycle analyses following the indicated treatment for 48 h in three different bladder cancer cell lines. DMSO served as solvent control. **(B)** Cyclin A, cyclin B1, cyclin D1, cyclin E, and CDK4 protein expression levels after gemcitabine and/or AZD6738 treatment were determined by western blot analysis in comparison with DMSO control in the bladder cancer cells.

gemcitabine and AZD6738 led to an increase in the fraction of sub-G1 in all the investigated cell lines after 48 h, compared with the untreated controls or each single treatment.

Western blot analysis showed that the G2–M-phase cyclins A and B1 were diminished upon combined treatment with gemcitabine and AZD6738 after 48 h; however, the G2–M-phase fraction did not decrease (Fig 2B). Similarly, the expression of G1-phase-related cyclin D1 and S-phase-related cyclin E decreased after 48 h of the combination treatment, especially in the T24 and UM-UC-3 cells. The combination of gemcitabine and AZD6738 for 48 h also suppressed CDK4 expression. Thus, the expression of cyclins was evidently perturbed, suggesting disturbances in cell cycle progression.

## Apoptosis induced by the combination of gemcitabine and AZD6738 in bladder cancer cells

To characterize the cellular effects of the drug combination in more detail, we investigated the induction of apoptosis. The number of early apoptotic cells, as determined by Annexin V staining, was enhanced after the combination treatment (Fig 3A, S4 Fig). Concordantly, the levels of cleaved PARP and active caspase 3 increased as per the western blot analysis following the combination treatment in all the investigated cell lines (Fig 3B). These apoptosis markers were not or only weakly elevated by either single agent treatment. In the J82 and UM-UC-3 cell lines, cleaved PARP levels increased after 24 h of treatment. Therefore, these results indicate that the combination of gemcitabine and AZD6738 efficiently induced apoptosis in all investigated bladder cancer cells.

## Inhibition of DNA damage repair by AZD6738

To characterize the anticancer mechanisms, we determined through western blotting whether gemcitabine and AZD6738 induced γH2A.X (phosphorylated histone H2A.X on Ser139), which is positively correlated with DSBs and has been utilized as a marker of DSBs [21] (Fig 4A). The exposure of bladder cancer cells to gemcitabine and AZD6738 elevated the γH2A.X levels after 24 and 48 h of treatment, indicating that AZD6738 disturbed gemcitabine-induced DNA damage repair. Next, we evaluated the expression of Rad51, which plays a major role in the homologous recombination repair of DNA. Rad51 expression was decreased in the T24

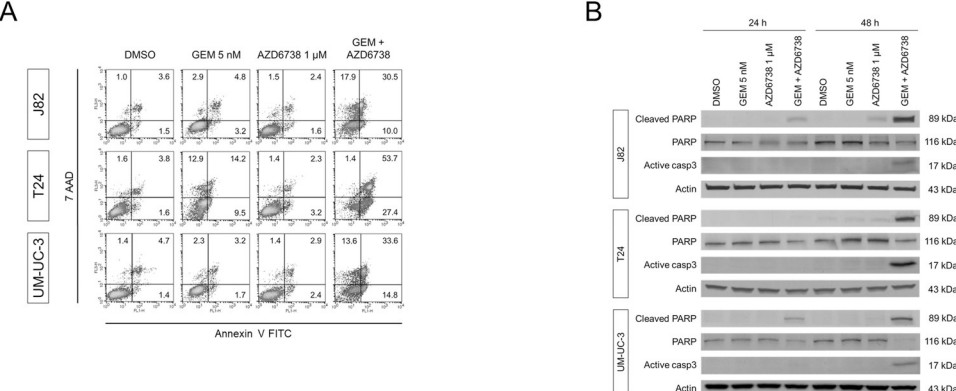

**Fig 3. Induction of apoptosis after treatment with gemcitabine and/or AZD6738. (A)** Flow cytometric analysis of bladder cancer cells with the indicated treatment after combined staining with Annexin V and 7-amino-actinomycin D (7AAD). Percentages of viable (lower left), early (lower right), or late (upper right) apoptotic and necrotic (upper left) cells subsequent to indicated treatments. **(B)** PARP cleavage and active caspase 3 48 h after treatment assessed by western blot analysis.

A 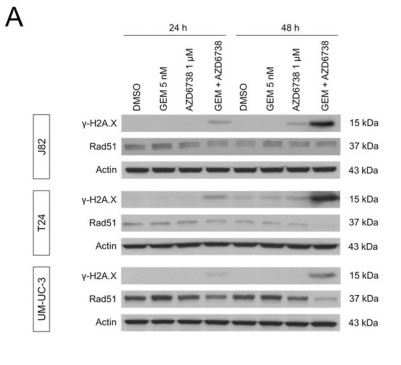

B 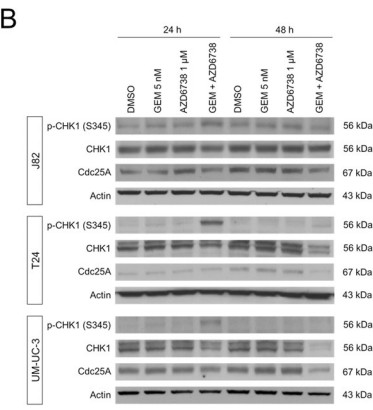

**Fig 4. Western blot analysis of checkpoint factors.** Whole cell lysates from bladder cancer cells treated with gemcitabine (5 nM) and/or AZD6738 (1 μM) for 24 or 48 hours were assayed for the indicated proteins or their phosphorylation. As loading controls, actin was stained on each blot.

and UM-UC-3 cells after 48 h of gemcitabine and AZD6738 combination treatment, suggesting that exposure to the combination treatment decreased homologous recombination activity and caused the accumulation of DNA damage.

To confirm that AZD6738 inhibits ATR in bladder cancer cells, CHK1 signaling was investigated (Fig 4B). The expression of ATR-mediated phosphorylation of CHK1 (Ser345 CHK1) was transiently elevated 24 h after treatment with the combination of gemcitabine and AZD6738, which was consistent with pronounced genotoxic stress. After 48 h, AZD6738 inhibited CHK1 phosphorylation and decreased the expression of the downstream transcriptional target cdc25A, which dephosphorylates cyclin-dependent kinases and regulates the cell cycle. The decreased expression of cdc25A after 48 h of treatment suggested that the activation of CHK signaling was inhibited by the co-administration of AZD6738 and gemcitabine.

## Discussion

Based on the data retrieved in this study, we found that the ATR inhibitor AZD6738 can enhance the cytotoxic effects induced by gemcitabine, a ribonucleotide reductase inhibitor, in bladder cancer cells. The results of our current study clearly demonstrate a significant synergistic effect of the gemcitabine and AZD6738 combination treatment on cell proliferation and prominent apoptotic cell death. In line with our results, we have previously demonstrated that the CHK1 inhibitor MK-8776 sensitizes bladder cancer cells to gemcitabine. This implies that inhibition of the ATR-CHK1 pathway enhances the cellular sensitivity of bladder cancer cells to gemcitabine. As bladder cancer cells are considerably resistant to apoptosis induction under many conditions, we are characterizing these cellular effects and molecular mechanisms using the ATR inhibitor AZD6738. These lines of evidence suggest that AZD6738 exerts a gemcitabine-sensitizing effect *in vitro* and may have clinical potential in combination with gemcitabine.

Cells are constantly exposed to a wide variety of genotoxic stressors. To overcome DNA damage, cells have evolved a complex mechanism termed the DNA damage response, comprising DNA repair and cell cycle checkpoint pathways [22]. DNA single-strand breaks are discontinuities in one strand of the DNA duplex, and they represent the most common type of DNA damage. Unrepaired single-strand breaks result in DNA replication stress and are converted into DSB during the S phase, resulting in genome instability [23]. The cytotoxic action of gemcitabine is related to its incorporation into DNA, causing cell cycle arrest in the S phase

[24], as demonstrated in Fig 2A. ATR and the downstream cell cycle checkpoint kinases are activated by gemcitabine-induced DNA damage, and AZD6738 selectively inhibits ATR and abrogates cell cycle arrest. Here, we found that the chemosensitizing ability of AZD6738 is associated with the abrogation of gemcitabine-induced cell cycle arrest and the promotion of DNA damage. In the current study, ATR inhibition by AZD6738 had a measurable impact on DSB repair kinetics after exposure to gemcitabine; nevertheless, ATR inhibition on its own had no measurable impact on DSB repair kinetics (Fig 4A). This suggests that ATR inhibition influences gemcitabine-induced DSB levels and repair kinetics. In particular, AZD6738 increased CHK1 phosphorylation at Ser345 (ATR-mediated CHK1 phosphorylation) and enhanced DSBs induced by gemcitabine after 24 h of treatment (Fig 4B). This is likely linked to its inhibitory effects on homologous recombination, the major form of DSB repair mechanism that depends on the presence of undamaged sister chromatids as a repair template in the S or G2 phases [25–27].

Clinical studies testing antitumor activity in combination with DNA-damaging agents are ongoing [28]. A phase I clinical trial using the ATR kinase inhibitor AZD6738 in combination with paclitaxel have shown that AZD6738 is well tolerated and exhibits antitumor activity in patients with advanced solid tumors [29]. Antitumor activity was also observed in patients with advanced cancer who had failed standard chemotherapy. However, to the best of our knowledge, there are no clinical trials using AZD6738 with gemcitabine. Previous preclinical studies have shown that AZD6738 can sensitize cells to gemcitabine via inhibition of gemcitabine-induced CHK1 activation, prevention of cell cycle arrest, and accumulation of restrained ribonucleotide reductase M2 [16]. Nevertheless, the exact mechanism by which AZD6738 affects bladder cancer cells in response to gemcitabine remains unknown. The *in vitro* experiments performed in the current study showed that the co-administration of AZD6738 with gemcitabine in the three bladder cancer cell lines resulted in persistent inhibitory effects on homologous recombination and persistent double-strand breaks for at least 48 h. The current study therefore suggests that the combined use of AZD6738 and gemcitabine may be more effective in patients with advanced bladder cancer.

Despite these important findings, our study has some limitations. First, our findings cannot simply be extended to all bladder cancer cells, despite our investigation of the effects of pharmacological ATR inhibition in three bladder cancer cell lines that represent bladder cancer heterogeneity. Further investigations should also identify whether the cytotoxic effect of the combination treatment is related to any cellular characteristics and thereby predict the response to ATR inhibitors, although we hypothesize that tumors with a defective DNA damage response are more likely to respond to ATR inhibition. Another limitation of this study is that the efficacy of the gemcitabine and AZD6738 combination has not been evaluated in animal models. We believe that studies in animal models (xenograft or carcinogen-induced) should be the next step. However, given the data from other cancer types, it is likely that the drug combination would be effective in suppressing tumor growth and would be well tolerated in animal models, without weight loss. The potentiation of gemcitabine effects by AZD6738 in xenografts from several other cancer types has been previously reported [16, 30]. The next step towards the application of our results in bladder cancer should therefore be animal experiments evaluating side effects and determining the optimal dosage.

## Conclusions

The ATR inhibitor AZD6738 enhanced gemcitabine activity in bladder cancer cells by inhibiting gemcitabine-induced DNA damage response. Thus, our study demonstrates the potential of agents that target the DNA replication stress response as a therapeutic strategy to treat

bladder cancer. As such, the current study provides a rationale for testing ATR inhibitors in combination with gemcitabine in patients with bladder cancer, particularly for patients with advanced and/or metastatic disease.

## Supporting information

**S1 Fig. Colony formation assay following treatment with gemcitabine (5 nM) and/or AZD6738 (1 μM) for 24 or 48 h.** DMSO was used as negative control. Bar graphs show the relative density of the cells at each treatment. $^*p < 0.05$, $^{**}p > 0.05$.
(TIF)

**S2 Fig. Morphology of the J82 and UM-UC-3 cells visualized by light microscopy with or without treatment with gemcitabine and AZD6738 at the indicated concentrations.** Scale bar: 100 μm.
(TIF)

**S3 Fig. Cell cycle analysis by flow cytometry following treatment with gemcitabine (5 nM) and/or AZD6738 (1 μM) for 24 or 48 h.** DMSO was used as negative control. (A) Bar graphs show the relative distribution of the cells at each phase of the cell cycle. (B) Bar graphs show the percentages of the cells in the sub-G1 fraction. $^*p < 0.05$, $^{**}p > 0.05$.
(TIF)

**S4 Fig. Flow cytometric analysis of bladder cancer cells treated with indicated conditions after combined staining with Annexin V and 7-AAD.** (A) The results are expressed as a percentage of early apoptotic cells, late apoptotic cells and necrotic cells. Bar graphs show the relative distribution of the cells at each quadrant. (B) Bar graphs show the percentages of apoptotic cells. $^*p < 0.05$.
(TIF)

**S5 Fig. Uncropped blots corresponding to Figs 2B, 3B, 4A and 4B.** Arrows indicate cropped bands. Note that the membranes were cut before probing.
(PDF)

## Author Contributions

**Conceptualization:** Makoto Isono.

**Data curation:** Makoto Isono, Kazuki Okubo.

**Formal analysis:** Makoto Isono, Akinori Sato.

**Investigation:** Makoto Isono.

**Methodology:** Makoto Isono, Takako Asano, Akinori Sato.

**Project administration:** Makoto Isono.

**Writing – original draft:** Makoto Isono.

**Writing – review & editing:** Makoto Isono.

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
