## [Decision Letter · Decision Letter 0]

12 Nov 2021

PONE-D-21-29989Ataxia telangiectasia and Rad3-related inhibition by AZD6738 enhances gemcitabine-induced cytotoxic effects in bladder cancer cellsPLOS ONE

Dear Dr. Isono,

Thank you for submitting your manuscript to PLOS ONE. After careful consideration, we feel that it has merit but does not fully meet PLOS ONE’s publication criteria as it currently stands. Therefore, we invite you to submit a revised version of the manuscript that addresses the points raised during the review process.

 You will note that all the reviewers have raised similar concerns that will be critical to address. Please make the needed experimental and textual changes as indicated. 

We look forward to receiving your revised manuscript.

Kind regards,

Robert W Sobol, PhD

Academic Editor

PLOS ONE

Journal Requirements:

Reviewers' comments:

Reviewer's Responses to Questions

**Comments to the Author**

1. Is the manuscript technically sound, and do the data support the conclusions?

Reviewer #1: Yes

Reviewer #2: Yes

Reviewer #3: Partly

2. Has the statistical analysis been performed appropriately and rigorously? 

Reviewer #1: No

Reviewer #2: Yes

Reviewer #3: I Don't Know

3. Have the authors made all data underlying the findings in their manuscript fully available?

Reviewer #1: Yes

Reviewer #2: Yes

Reviewer #3: No

4. Is the manuscript presented in an intelligible fashion and written in standard English?

Reviewer #1: Yes

Reviewer #2: Yes

Reviewer #3: Yes

5. Review Comments to the Author

Reviewer #1: This article has a straightforward take home message that AZD6738 can increase the sensitivity of bladder cancer cells to gemcitabine.

The data seem strong. However, the quality of the images is very poor and most of them are not readable at all.

One other issue is a complete lack of statical analyses on all the figures although some error bars are present. Appropriate statics should be included for significance of the data.

More specifically:

Fig1: authors mention that the number of morpholigally abnormal cells increase. Yet there is no quantification provided, only images. Either this statement has to be removed or quantifications have to be performed and provided with appropriate statiscal analyses.

Sup Fig 3: the legend such as “lower left” etc...does not seem appropriate. Authors may consider naming them viable, early, late directly.

Finally, can the authors comment on why CHk1 phosphorylation on S345 is still happening while ATR is inhibited. It seems counter-intuitive and should be further discussed.

Reviewer #2: Overall: Isono et al investigated the synergistic use of gemcitabine and ATR inhibitor AZD6738 to treat bladder cancer. In vitro studies using 3 bladder cancer cell lines demonstrated clear promise for this combination, as results indicated induction of apoptosis, as well as inhibition of cell viability, colony formation, and cell cycle progression to a greater extent than was seen with either gemcitabine or AZD6738 alone. Moreover, Western blot analyses of checkpoint factors provided evidence that AZD6738 works by inhibiting the repair of gemcitabine-induced double stranded breaks. The study is well conduced, and the results are well described and support the conclusions. While this study is an important and vital step towards understanding the potential for this drug combination in bladder cancer, enthusiasm is a bit diminished due to previous works demonstrating the success of this combination, and its gemcitabine sensitizing mechanism, in other cancer types.

Minor:

The last sentence on the first page of the introduction loses clarity starting with “… or resected DBSs recruits and activates…”

Reviewer #3: In this study, the authors report that inhibition of ATR kinase by AZD6738 can sensitize bladder cancer cells to gemcitabine therapy. The design of the study is straightforward, including analyses of the combined effects of gemcitabine and AZD6738 on cell viability, clonogenicity, cell cycle, apoptosis, DNA damage, DNA repair and DNA damage checkpoint signaling. Three bladder cancer cells are included in the experiments. While the similar synergistic effects have been observed for many human cancer cell lines, this study add bladder cancer cell lines to the list. Overall, this is an incremental study to extend our understanding on the significance of ATR inhibition in sensitizing cancer cells to gemcitabine.

1. Since the quality of all figure images is extremely poor and the images are at very low resolution, the data cannot be read by this reviewer to make proper judgements on the quality of data and how the experiments were designed and performed. Thus, no comments can be made.

2. AZD6738 inhibits both ATR and ATM. There are more specific ATR inhibitors such as VX970 or BAY 1895344, which should be used.

6. PLOS authors have the option to publish the peer review history of their article (what does this mean?). If published, this will include your full peer review and any attached files.

Reviewer #1: No

Reviewer #2: No

Reviewer #3: No

---

## [Author Response · Author response to Decision Letter 0]

19 Jan 2022

Response to Reviewers

Reviewer #1: 

This article has a straightforward take home message that AZD6738 can increase the sensitivity of bladder cancer cells to gemcitabine.

The data seem strong. However, the quality of the images is very poor and most of them are not readable at all.

Response: Thank you for the valuable suggestion. We have corrected the figures.

One other issue is a complete lack of statical analyses on all the figures although some error bars are present. Appropriate statics should be included for significance of the data.

Response: Thank you for the valuable suggestion. We think your comment makes a valid point. However, we do not think it is essential to perform statistical analyses in the current experiments because of sample size. Supplementary figures have been added to show the results quantitatively.

More specifically:

Fig1: authors mention that the number of morphologically abnormal cells increase. Yet there is no quantification provided, only images. Either this statement has to be removed or quantifications have to be performed and provided with appropriate statistical analyses.

Response: Thank you for the valuable suggestion. This statement has been removed.

Sup Fig 3: the legend such as “lower left” etc...does not seem appropriate. Authors may consider naming them viable, early, late directly.

Response: Thank you for the valuable suggestion. The text of the manuscript and the figure have been corrected.

Finally, can the authors comment on why CHk1 phosphorylation on S345 is still happening while ATR is inhibited. It seems counter-intuitive and should be further discussed.

Response: Thank you for the valuable suggestion. CHK1 kinase acts downstream of ATR kinase and also ATM kinase. Activation of CHK1 at Ser 345 occurs in response to blocked DNA replication and certain forms of genotoxic stress. ‘On target’ inhibition of ATR by AZD6738 may induce the potent activation of ATM by combined administration of gemcitabine and the cells attempt to compensate (Wallez Y et al. The ATR inhibitor AZD6738 synergizes with gemcitabine in vitro and in vivo to induce pancreatic ductal adenocarcinoma regression. Mol Cancer Ther. 2018; 17(8)).

Reviewer #2: 

Overall: Isono et al investigated the synergistic use of gemcitabine and ATR inhibitor AZD6738 to treat bladder cancer. In vitro studies using 3 bladder cancer cell lines demonstrated clear promise for this combination, as results indicated induction of apoptosis, as well as inhibition of cell viability, colony formation, and cell cycle progression to a greater extent than was seen with either gemcitabine or AZD6738 alone. Moreover, Western blot analyses of checkpoint factors provided evidence that AZD6738 works by inhibiting the repair of gemcitabine-induced double stranded breaks. The study is well conduced, and the results are well described and support the conclusions. While this study is an important and vital step towards understanding the potential for this drug combination in bladder cancer, enthusiasm is a bit diminished due to previous works demonstrating the success of this combination, and its gemcitabine sensitizing mechanism, in other cancer types.

Minor:

The last sentence on the first page of the introduction loses clarity starting with “… or resected DBSs recruits and activates…”

Response: Thank you for the valuable suggestion. The text of the manuscript has been corrected.

Reviewer #3: 

In this study, the authors report that inhibition of ATR kinase by AZD6738 can sensitize bladder cancer cells to gemcitabine therapy. The design of the study is straightforward, including analyses of the combined effects of gemcitabine and AZD6738 on cell viability, clonogenicity, cell cycle, apoptosis, DNA damage, DNA repair and DNA damage checkpoint signaling. Three bladder cancer cells are included in the experiments. While the similar synergistic effects have been observed for many human cancer cell lines, this study add bladder cancer cell lines to the list. Overall, this is an incremental study to extend our understanding on the significance of ATR inhibition in sensitizing cancer cells to gemcitabine.

1. Since the quality of all figure images is extremely poor and the images are at very low resolution, the data cannot be read by this reviewer to make proper judgements on the quality of data and how the experiments were designed and performed. Thus, no comments can be made.

Response: Thank you for the valuable suggestion. We have corrected the figures.

2. AZD6738 inhibits both ATR and ATM. There are more specific ATR inhibitors such as VX970 or BAY 1895344, which should be used.

Response: Thank you for the valuable suggestion. The reason I have chosen AZD6738 in this study is that it is the first bioavailable ATR kinase inhibitor described, and was shown to enhance the therapeutic efficacy of gemcitabine in xenograft models in other cancer types. Also, it is orally active in clinical trials. I am very interested in using more specific ATR inhibitors such as VX970 or BAY 1895344, and it is the next step. Thank you.

---

## [Decision Letter · Decision Letter 1]

22 Mar 2022

Ataxia telangiectasia and Rad3-related inhibition by AZD6738 enhances gemcitabine-induced cytotoxic effects in bladder cancer cells

PONE-D-21-29989R1

Dear Dr. Isono,

We’re pleased to inform you that your manuscript has been judged scientifically suitable for publication and will be formally accepted for publication once it meets all outstanding technical requirements.

Kind regards,

Robert W Sobol, PhD

Academic Editor

PLOS ONE

Reviewers' comments:

Reviewer's Responses to Questions

**Comments to the Author**

1. If the authors have adequately addressed your comments raised in a previous round of review and you feel that this manuscript is now acceptable for publication, you may indicate that here to bypass the “Comments to the Author” section, enter your conflict of interest statement in the “Confidential to Editor” section, and submit your "Accept" recommendation.

Reviewer #2: All comments have been addressed

Reviewer #3: (No Response)

2. Is the manuscript technically sound, and do the data support the conclusions?

Reviewer #2: Yes

Reviewer #3: (No Response)

3. Has the statistical analysis been performed appropriately and rigorously? 

Reviewer #2: I Don't Know

Reviewer #3: (No Response)

4. Have the authors made all data underlying the findings in their manuscript fully available?

Reviewer #2: Yes

Reviewer #3: (No Response)

5. Is the manuscript presented in an intelligible fashion and written in standard English?

Reviewer #2: Yes

Reviewer #3: (No Response)

6. Review Comments to the Author

Reviewer #2: (No Response)

Reviewer #3: I was disappointed that the authors have made no improvement on the display quality of the figures. All the figures are basically the same as in the original manuscript and are unreadable as all data labels look fuzzy. There is no way reviewers can read and review this manuscript. I wonder if the authors submitted a wrong version of their revised manuscript.

7. PLOS authors have the option to publish the peer review history of their article (what does this mean?). If published, this will include your full peer review and any attached files.

Reviewer #2: No

Reviewer #3: No

---

## [Editor Report · Acceptance letter]

1 Apr 2022

PONE-D-21-29989R1 

Ataxia telangiectasia and Rad3-related inhibition by AZD6738 enhances gemcitabine-induced cytotoxic effects in bladder cancer cells 

Dear Dr. Isono:

I'm pleased to inform you that your manuscript has been deemed suitable for publication in PLOS ONE. Congratulations! Your manuscript is now with our production department. 

Kind regards, 

on behalf of

Dr. Robert W Sobol 

Academic Editor

PLOS ONE